# SSL-SLR: Self-Supervised Representation Learning for Sign Language Recognition

**Ariel Basso Madjoukeng**                                    *arielbassodev@gmail.com*
*University of Namur*

**Jérome Fink**                                               *jerome.fink@unamur.be*
*University of Namur*

**Pierre Poitier**                                            *pierre.poitier@unamur.be*
*University of Namur*

**Edith Bélise Kenmogne**                                     *ebkenmogne@gmail.com*
*University of Dschang*

**Benoit Frenay**                                             *benoit.frenay@unamur.be*
*University of Namur*

**Reviewed on OpenReview:** *https://openreview.net/forum?id=buTZkTXijy*

## Abstract

Sign language recognition (SLR) is a machine learning task aiming to identify signs in videos. Due to the scarcity of annotated data, unsupervised methods like contrastive learning have become promising in this field. They learn meaningful representations by pulling positive pairs (two augmented versions of the same instance) closer and pushing negative pairs (different from the positive pairs) apart. In SLR, only certain parts of the sign videos provide information that is truly useful for their recognition. Applying contrastive methods to SLR raises two issues: (i) contrastive learning methods treat all parts of a video in the same way, without taking into account the relevance of certain parts over others; (ii) shared movements between different signs make negative pairs highly similar, complicating sign discrimination. These issues lead to learning non-discriminative features for sign recognition and poor results in downstream tasks. In response, this paper proposes a self-supervised learning framework designed to learn meaningful representations for SLR. This framework consists of two key components designed to work together: (i) a new self-supervised approach with free-negative pairs; (ii) a new data augmentation technique. This approach shows a considerable gain in accuracy compared to several contrastive and self-supervised methods, across linear evaluation, semi-supervised learning, and transferability between sign languages.

## 1 Introduction

Sign language recognition is a challenging task that involves identifying signs in videos. Depending on the data format, there exist video-based SLR and image-based SLR. Sign language recognition with machine learning is rapidly expanding, but it faces several challenges, particularly in the acquisition of annotated datasets. Indeed, collecting and annotating sign language data requires linguistic expertise that is difficult to find, it is costly and time-consuming (De Coster et al., 2024). As an example, annotating 1 hour of sign videos takes about 100 hours (Renz et al., 2021). For this reason, sign language suffers from a scarcity of annotated data.

Recent studies have turned to unsupervised methods such as contrastive representation learning. It offers a promising solution enabling deep learning models to train without supervision by leveraging data augmentations. Once pre-trained on unannotated sign language data, the model can then be fine-tuned on less available data and achieve good results. Most contrastive approaches generate positive and negative pairs (Chen et al., 2020a; He et al., 2020) then learn similarities and dissimilarities from them. A positive pair is usually created by applying an augmentation to an instance, while negative pairs consist of different instances from the dataset. As an example, for an image, a positive pair consists of the image with a new color distortion and the rotated image.

Li et al. (2020b) show that contrastive approaches can generate negative pairs with similar semantic characteristics, resulting in a poorly discriminated latent space. This issue is amplified in sign language where distinct signs may share similar movements (Vogler & Metaxas, 2004) and there are several signs that share similar hand shapes and movements but convey different meanings (Zuo et al., 2023). Approaches without negative (e.g., BYOL (Grill et al., 2020)) pairs often require additional components (encoder, or other), increasing the complexity of the model. Additionally, sign language videos include repositioning (hand adjustments following a sign) and coarticulation (transitory motions between signs) (Poitier et al., 2024). These movements are generally present in sign language data, but they are not useful for sign discrimination. Thus, not all parts of the signs are relevant for their identification. Contrastive learning approaches are designed to learn invariant representations by leveraging data augmentation. Thus, using contrastive approaches, all representations even those irrelevant for identification are learned. This results in low accuracy during the linear evaluation, poorly discriminated embedding space and limited transferability of learned representations. Methods that focus on local aspects (Xie et al., 2021; He et al., 2020; Bardes et al., 2022) of images are not suitable for this task, as they only concentrate on individual elements, rather than entire segments of the video recording.

In response to the above challenges, this paper proposes a self-supervised framework consisting of two key components: a new self-supervised approach and a new augmentation method. The self-supervised method is designed to be invariant to augmentations by producing similar representations for a sign and its augmented variants. While the proposed data augmentation generates positive pairs by degrading the non-relevant parts of the signs, enabling the self-supervised method to become invariant to those non-relevant parts. The proposed method removes the need for negative pairs, additional encoders, or clustering mechanisms and surpasses several well-known contrastive and self-supervised approaches on several SLR tasks. The results show first an improvement in the accuracy and quality of representations learned by the most popular contrastive architectures, such as SimCLR, MoCo, SimSiam, and BYOL, across several sign language datasets. Second, the achievement of state-of-the-art results on several datasets.

The rest of this paper is organized as follows: Section 2 presents contrastive learning and several popular methods generally used in SLR. Section 3 presents previous works in SLR; Section 4 shows that not all parts of a sign language video are equally relevant for the recognition; Section 5 presents the proposed approach; Section 6 presents experiments and the discussion; Section 7 presents the ablation of the approach while Section 8 concludes and presents several further perspectives.

## 2 Contrastive Representation Learning

Contrastive learning enables deep learning models to learn relevant representations without annotations. Many contrastive learning methods exist, each with its own particularities; SimCLR (Chen et al., 2020a) is one of the most popular approaches. At each step, it generates positive pairs through data augmentations, then maximizes the similarity between positive pairs while minimizing the similarity between positive pairs and the negative others. However, in this approach, positive and negative pairs are restricted to the instances within the batch, which limits the learned diversity. MoCo v2 (Chen et al., 2020b) addresses this limitation by introducing a queue to store the batch of the previous iteration and use two encoders. In SimCLR and MoCo v2, many negative pairs that often share similar characteristics are generated. This complicates the discrimination and often yields a poorly discriminated embedding space. In response, Li et al. (2020b) introduced the prototypical contrastive learning (PCL). It follows the MoCo approach, but at each step, it uses the $k$-means clustering (MacQueen, 1967) on instances from the momentum encoder. The issue with

the PCL is the use of a clustering function that requires a number of clusters. Modern contrastive methods such as BYOL or SimSiam avoid this issue by using only positive pairs. BYOL uses two encoders (the online and the target), at each step it generates positive pairs which are then passed through the encoders. The online encoder is trained to predict the target encoder. SimSiam for its part employs a straightforward Siamese architecture (Bromley et al., 1993) with one encoder to learn representations from augmentations. These approaches are generally designed and validated for image classification and segmentation tasks. They have been used in the SLR and have yielded interesting results (Kothadiya et al., 2023; Madjoukeng et al., 2025a;b). The next section presents previous works in SLR.

## 3 Sign Language Recognition

Similar to spoken languages, there exist many sign languages that are different from one region to another. Several SLR approaches have been developed for various sign languages. For the Argentinian sign language, Masood et al. (2018) proposed a two-level architecture (CNN and RNN) for the recognition of 46 different signs and achieved 95.2% accuracy. This accuracy can be attributed to the limited number of classes. Marais et al. (2022) showed that an InceptionV3-GRU model (Szegedy et al., 2016) trained from scratch achieved 74.22% accuracy on a vocabulary of 64 different signs (Ronchetti et al., 2023). In the same sign language, Alyami et al. (2024) benchmarked several architectures and found that the best result (98.25% accuracy) was achieved with a transformer model.

For the French Belgian sign language (LSFB), Fink et al. (2021; 2023) introduced a comprehensive dataset containing 4,567 distinct signs, thereby providing a valuable resource for the development and evaluation of sign language recognition systems associated with this region. Leveraging this dataset, they trained a Vision Transformer (Dosovitskiy et al., 2020) architecture from scratch. Despite the complexity and variability inherent in this dataset, their model achieved 54.4% accuracy on a subset of 700 signs.

For the American sign language, Li et al. (2020a) proposed a pose-based Temporal Graph Convolutional Neural Network (Pose-TGCN) for the recognition of the signs of this language. They achieved 55.43 % accuracy on a subset of 100 signs of Word Level American sign language (WLASL). Tunga et al. (2021) proposed GCN-BERT, a framework for pose-based SLR that combines a Graph Convolutional Network (GCN) and BERT and achieved 60.15% accuracy. Hu et al. (2021) presented SignBERT, a large-scale model pretrained on a vast amount of data using a pretext task of masked visual token reconstruction. After pre-training and fine-tuning their model, they achieved an accuracy of 76.36% on the same subset (100 signs). Zhao et al. (2023) proposed BEST (BERT Pre-training for Sign Language Recognition with Coupling Tokenization), a framework pre-trained on a large amount of data with a pretext task of reconstructing the masked unit (left hand, right hand, etc.). With their approach, they obtained 77.91% accuracy on the same subset. Hu et al. (2023) proposed SignBERT+, an enhancement of SignBERT which incorporates a model-aware hand prior. This improved model achieved 79.84% accuracy. More recently, Jiang et al. (2024) proposed Sign-CLIP, a large model trained on 500,000 videos and text descriptions from 44 different sign language corpora. The architecture involves two encoders: a video encoder and a text encoder. During the training, both encoders are optimized jointly using a contrastive loss to align video and text representations in a shared latent space. With this approach, the model achieved 46% accuracy on the ASL Citizen dataset. Wong et al. (2025) proposed SignRep, a self-supervised learning framework for sign language recognition. Pretrained on the YouTube-SL dataset (Tanzer & Zhang, 2024) (a large corpus containing more than 3000 hours of videos across 25 sign languages), they achieved 49.95% accuracy on this dataset during fine-tuning. Other works (Gueuwou et al., 2025a;b) have also leveraged the YouTube-SL dataset to pretrain models for signs recognition in this language.

For the Greek sign language, Adaloglou et al. (2021) conducted an extensive benchmark of several deep learning architectures to determine the most effective model. In their experiments, the best performance was achieved using a hybrid architecture combining an Inception-3D with a Bidirectional Long Short-Term Memory (BLSTM) module. This design allows the model to benefit both from spatiotemporal feature extraction (via Inception-3D) and from temporal sequence modeling (via BLSTM). With their architecture, they achieved an accuracy of 89.74% on a vocabulary of 310 isolated signs. Recently, on this dataset, Papadimitriou et al. (2024) introduced a multimodal framework that leverages both appearance-based information

and skeleton-based information. For the appearance features, they used a ResNet2+1D network, while for skeleton sequences, they employed a spatio-temporal graph convolutional network. When applied to this dataset, the model achieved 96.25% accuracy.

Worldwide, there exist over 150 sign languages (Bilge et al., 2024), but only a few have enough annotated data to train deep learning models effectively. Existing models ((Hu et al., 2021; 2023; Zhao et al., 2023; Jiang et al., 2024)) rely on huge amounts of data from multiple sign languages to pre-train models, which are then fine-tuned with few annotated data. However, this presents some challenges, indeed pre-training requires large datasets (e.g., 500,000 videos for SignCLIP) from diverse sources (SignBERT, SignBERT+, etc.), which are sometimes not accessible and require considerable computational resources; additionally, they often lack sign-specific features (Wong et al., 2025). For an effective solution to data scarcity in SLR, instead of pre-training models on vast amounts of data from diverse sources, recent studies are focusing on pre-training the models on unannotated data from a sign language or leverage self-supervised learning approaches (Kothadiya et al., 2023; Madjoukeng et al., 2025a;b). In this context, Madjoukeng et al. (2025a) leveraged contrastive learning techniques based on data augmentation (SimCLR, MoCo, SimSiam, etc.) to learn representations from unannotated sign language data. They pre-trained and fine-tuned several contrastive models on data from a specific sign language without requiring vast amounts of data from diverse sources. Conducted on image-based SLR, this work has yielded remarkable results. By enabling the training of models in an unsupervised manner on a sign language, contrastive approaches provide an innovative solution to address the challenges of scarce annotated data in sign language. Madjoukeng et al. (2025b) show that by using contrastive learning approaches in image-based SLR, it sometimes occurs that the learned representations are not aligned with the sign in the image. They show that by enabling contrastive models to be more focused on the signs in the images, the accuracy and faithfulness of the contrastive models increase. As in image-based SLR, where the sign occupies only a portion of the image, we observe that even for video-based SLR, not all frames in a sequence are equally relevant for sign identification. The next section demonstrates that for sign identification, only certain frames are truly useful.

## 4    Not All Parts Of A Sign Are Relevant For Its Recognition

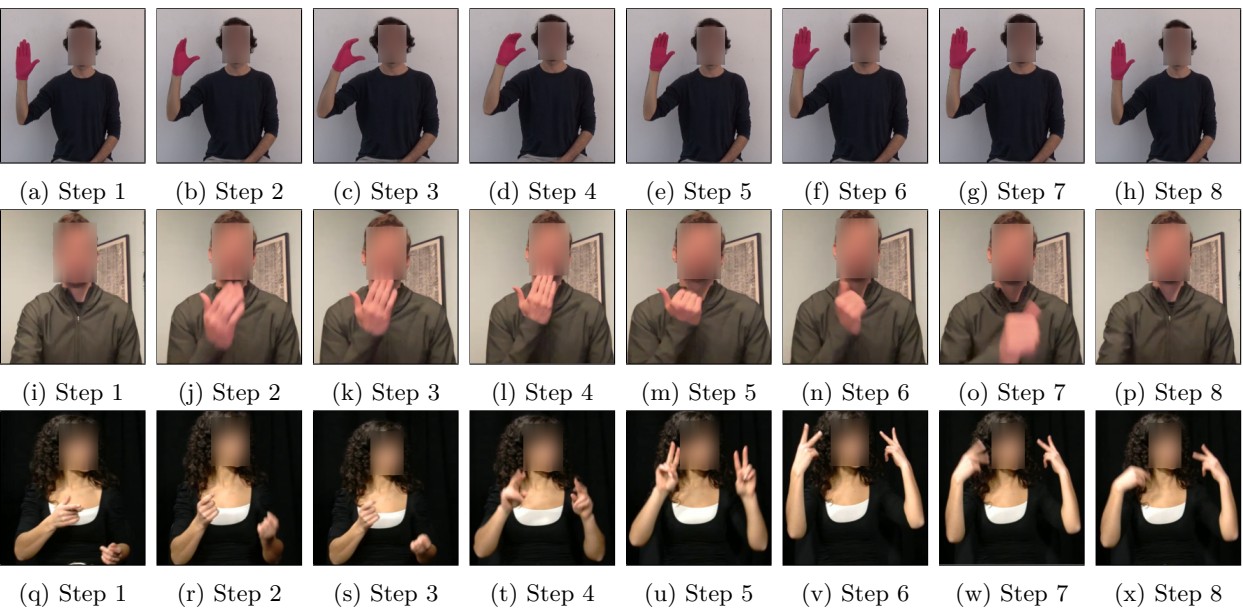

Figure 1: Examples of sign steps across three different datasets: LSA (Ronchetti et al., 2023) (first row), ASL (Desai et al., 2023) (second row), and LSFB (Fink et al., 2021) (third row). For privacy reasons, signers' faces have been blurred.

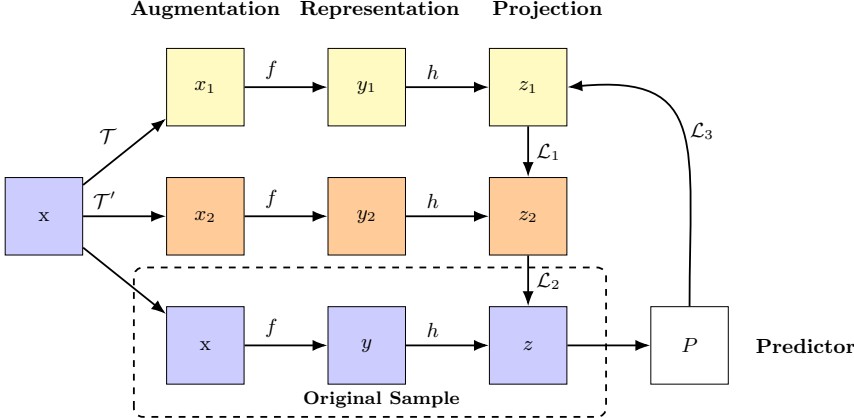

Figure 2: SL-FPN architecture: A sign and its augmented variants are passed through an encoder. SL-FPN optimizes three objectives: (1) minimizing the distance between representations of the two augmented variants; (2) minimizing the distance between representations of one augmented variant and the original instance; and (3) minimizing the distance between representations of the original sample and the other augmented variant using a predictor with a stop-gradient operator.

Datasets for sign languages are generally created in three ways. In some cases, signers are filmed in a studio performing isolated signs. In other cases, isolated signs are segmented from sign language videos and then annotated (e.g., LSFB). In other instances, the signers self-record the signs in a video with their equipment (e.g., ASL Citizen). In all scenarios, several issues occur: for signs recorded in a studio, repositioning movements appear. For signs resulting from segmentation due to segmentation errors, the beginning or end of a given sign often ends up in another sign. For the signs recorded by the signers themselves, at the beginning of the signs, the signers often turn on the camera, then turn it off after the signs. This leads to movements that are not truly useful for a model for an efficient and realistic discrimination. These movements are neither the essence, nor the frames on which a model should base for the signs discrimination.

To illustrate these movements, Figure 1 presents one instance of signs: *Opaque* (first), *Sweet* (second), *Also* (third), randomly selected from the LSA, LSFB, and ASL Citizen datasets. For each of these signs steps, several elements emerge: (i) at the start of the signs, the frame sequences are continuous and follow a consistent direction; (ii) at the end of each sign, repositioning movements are present (ASL Citizen) or sometimes preparing for the start of another sign (LSFB). This illustrates that, to identify a sign neither the initial position nor repositioning to the final position should matter, but rather what happens in between. To learn useful representations, models should focus more on the relevant parts of signs rather than learning representations that are irrelevant for sign identification. The next section presents an efficient self-supervised framework that leverages this argument to produce better representations for SLR.

## 5 Self-Supervised Framework For Sign Language Recognition (SSL-SLR)

This section presents the proposed self-supervised framework called SSL-SLR. This framework consists of a new self-supervised approach and a novel data augmentation. The first part of this section presents the self-supervised approach and the second the proposed data augmentation.

### 5.1 A novel self-supervised learning approach with free negative pairs: *SL-FPN*

The proposed SL-FPN aims to eliminate the need for negative pairs, additional clustering functions, or supplementary encoders that increase model complexity, while achieving higher accuracy than other negative-free methods. Generally, self-supervised methods only use either positive pairs or positive and negative pairs without leveraging the original instance. However, since both positive pairs and the original instance represent the same concept (i.e., the same sign in this case), it can be beneficial to leverage them simultaneously

during the learning process. This approach leverages both positive pairs and the original instance to enhance training. Figure 2 presents the proposed architecture. For an input $x$, two augmented versions are generated by randomly applying two different augmentations $\mathcal{T}$ and $\mathcal{T}'$ from an augmentation set. Then, the obtained versions $x_1$, $x_2$ and the original instance $x$, are each passed through an encoder $f$ and a projection head $h$. The representations $z_1 = h(f(x_1))$, $z_2 = h(f(x_2))$, and $z = h(f(x))$ are thus obtained. The goal of SL-FPN is to generate highly similar representations for an instance and its augmented versions. To achieve this, it minimizes the difference between the representations of an input and its augmented counterparts, typically using the Mean Squared Error (MSE) as the loss function. The smaller this difference, the better SL-FPN can produce closely aligned representations in the embedding space for an instance and its augmented variants. As shown in Figure 2, the SL-FPN loss is threefold and consists of: (i) an MSE between the representation of positive pairs ($\mathcal{L}_1$); (ii) an MSE between the representation of one of the positive pairs and the original instance ($\mathcal{L}_2$); (iii) an MSE between the output of the predictor $P$ and the representation of an augmented version. The stop-gradient operator defined as $sg(z_1) = z_1$ is used to consider one representation as constant and prevent gradient propagation along this representation. Like SimSiam or BYOL, it is used as an asymmetrical component, breaking the symmetry between the branches. Hence, in the SL-FPN architecture, the asymmetry is ensured by $\mathcal{L}_3$; without it, SL-FPN does not benefit from any asymmetric component. For $n$-sized embeddings, the final loss is given by:

$$\mathcal{L}_1 = \frac{1}{n}\|z_1 - z_2\|^2, \quad \mathcal{L}_2 = \frac{1}{n}\|z - z_2\|^2, \quad \mathcal{L}_3 = \frac{1}{n}\|P(z) - \text{sg}(z_1)\|^2, \quad \mathcal{L} = \mathcal{L}_1 + \mathcal{L}_2 + \mathcal{L}_3. \tag{1}$$

The training objective aims to minimize $\mathcal{L}(\theta)$, with $\theta$ the model parameters. For a set of $N$ samples and $\ell$ the MSE loss, $\mathcal{L}(\theta)$ is defined as:

$$\mathcal{L}(\theta) = \frac{1}{N}\sum_{i=1}^{N}\left[\ell(f_\theta(x_1^i), f_\theta(x_2^i)) + \ell(f_\theta(x^i), f_\theta(x_2^i)) + \ell(P(f_\theta(x^i)), \text{sg}(f_\theta(x_1^i)))\right] \tag{2}$$

By minimizing Equation 2, SL-FPN generates the closest representations for a sign and its augmented versions in the embedding space. During our experiments, we assigned a weight to each term of the objective function (equation 2), but this did not significantly affect the results. Therefore, for simplicity, we did not include any penalty term. The main difference between SL-FPN and other self-supervised methods such as BYOL (which uses two branches and two encoders) or SimSiam (which uses two branches and a predictor), lies in the fact that SL-FPN uses three branches, a single encoder and a predictor. The novelty here does not simply consist of using three branches, but also lies in the way that they are combined during training. Section 7 shows the effect of permuting the order of the three inputs on accuracy.

In representation learning without negative pairs, representation collapse often occurs. It occurs when the model produces the same representations for all input instances. To avoid this issue, several methods exist. Wu et al. (2024) show that layer normalization and skip connections in a transformer encoder can mitigate collapse. Chen et al. (2020a), and Grill et al. (2020) highlight the importance of stop-gradient mechanisms coupled with a predictor to prevent collapse in self-supervised architectures like SimSiam and BYOL. Hence, SL-FPN incorporates these components in its architecture to avoid the collapse solution. The ablation study in Section 7 shows the impact of each of these components in avoiding collapse in the SL-FPN. This approach can be applied to a wide range of computer vision tasks and can yield competitive results. In particular, it can be useful in situations where there is a risk of semantic inconsistency between positive pairs. This is a well-known issue in contrastive learning, where positive pairs may no longer share the same semantic information (Guo & Shi, 2023). In such cases, by leveraging the original instance, SL-FPN can prove to be very effective unlike other self-supervised approaches.

With the proposed SL-FPN architecture alone, the challenge of ensuring the relevance of the learned representations remains. Indeed, there is no guarantee that it will focus on features truly discriminative for sign identification. To address this issue, the next section introduces a new data augmentation strategy designed to help SL-FPN focus on the most relevant parts of the signs.

## 5.2 A new augmentation method

In contrastive learning, the augmented variants are generally obtained by applying augmentations to all parts of a sequence. By learning to produce similar representations between a sign and its augmented variant, contrastive learning methods aim to build an invariant representation from the augmentations. Madjoukeng et al. (2025b) show that, for image-based SLR, applying augmentations while preserving the region of interest (sign in the image) helps contrastive models to better focus on it and improve the performance in downstream tasks. Hence, in the same way, the proposed augmentation method aims to generate positive pairs by preserving the parts of the signs that are more discriminant for their identification. For image-based SLR, the region of interest can be identified using segmentation methods such as Mask R-CNN (He et al., 2017) or others. However, video-based SLR faces a major challenge: there is no established method to identify the frames that are relevant for sign recognition. To fill this gap, the first step of the proposed augmentation is to determine the relevant frames for the signs identification.

**Determining the relevant frames for signs identification**

In Section 4, it was shown that due to certain movements (repositioning, coarticulation, etc.), the information truly useful for discriminating signs is not found throughout the entire sequence. The fundamental question is therefore to determine where these discriminative parts begin and end within the sequence. The main objective of this step is to determine the *boundary importance*. It is the boundary that marks when frames become and stop being relevant for sign identification. This step involves identifying (i) the point in the sequence from which frames contain sufficient discriminative information ($k_s^*$); and (ii) the point until which these frames remain useful before losing their discriminative ability ($k_e^*$). To date, to the best of our knowledge, no standard method exists in the literature to locate these discriminative regions within a sequence. In this research, we propose an intuitive approach based on a contrastive algorithm with a transformer as backbone, combined with a temporal augmentation technique, denoted $\pi$.

A sign is a sequence of frames starting from an initial frame $k_s$ to a final frame $k_e$. The objective is to determine the optimal frames $k_s^*$ and $k_e^*$, which represent the *boundary importance*. Transformer-based architectures with positional encoding are particularly sensitive to the positioning of elements within the sequence. Contrastive methods generate positive pairs through augmentations and learn representations that are invariant to these augmentations. Thus, a contrastive approach equipped with a transformer encoder offers two key properties: (i) invariance to augmentations, inherited from the contrastive loss; (ii) sensitivity to the order of elements in the sequence, induced by the transformer positional encoding. Moreover, it is well established that the performance of contrastive models in downstream tasks (SLR in this case) strongly depends on the quality of the learned representations (Tian et al., 2020; Roschewitz et al., 2024). The detection of relevant frames therefore relies on the combination of these two properties and on the exploitation of this information.

The idea is to use a contrastive algorithm to progressively degrade a sign, first from the first to the last frame (to determine the frame $k_s^*$), then from the last to the first frame (to determine $k_e^*$). The impact of each degradation is then evaluated on linear evaluation. If the model becomes invariant to frames necessary for identifying a sign, its performance during linear evaluation will be low. Conversely, if it learns to be invariant to irrelevant frames, it will focus more on informative parts, leading to better downstream performance. For the degradation, several augmentations can be applied. Due to its capacity of modifying the temporal order of sequences while preserving the local distribution of the frames, temporal permutation is chosen. Additionally, as a transformer encoder with a positional encoding is sensitive to the positioning of each element in the sequence, temporal permutation will induce diversity between the positive pairs from the point of view of the transformer encoder. Contrary to augmentations such as rotations, Gaussian blur, flips that are applied at the local frame level.

Algorithm 1 presents an overview of this method. It takes as input a dataset $\mathcal{D}$, a contrastive algorithm $\mathcal{C}$, two augmentations $\pi_1$ and $\pi_2$ (typically two random permutations). Its goal is to determine $k_s^*$ and $k_e^*$ (boundary importance) whose permutation maximizes the classification accuracy of $\mathcal{C}$. For $k_s^*$, the algorithm evaluates the accuracy after permuting the first $k_s$ frames, increasing $k_s$ as long as accuracy increases. For $k_e^*$, it follows a symmetric procedure on the last $k_e$ frames. The `SegmentEval` function applies $\pi_1$ and $\pi_2$ to

the selected segment, inserts the modified segments back into the sequence, and measures accuracy via $\mathcal{C}$. The process stops for each parameter when further changes no longer improve accuracy and returns $(k_s^*, k_e^*)$ as the optimal lengths before stagnation. For datasets containing long sequences of signs, instead of iterating over all frames, the evaluation can proceed in frame steps (e.g., every $p$ frame) which reduces computational cost.

After determining the boundary importance, the proposed augmentation consists of generating positive pairs by applying augmentation on the first $k_s^*$ and on the last $k_e^*$ to help the contrastive model to be more focused on the relevant part of the sign. The next part of this section presents two case studies as proof of concept for determining $k_s^*$ and $k_e^*$: one on the GSL and the other on the LSFB dataset.

**Proof of concept: a study on the GSL and LSFB datasets**

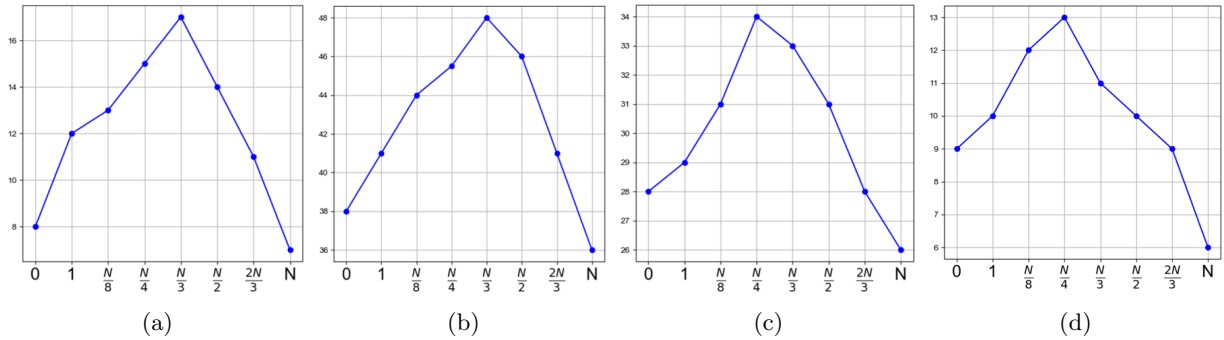

(a)         (b)         (c)         (d)

Figure 3: Variation of the accuracy during linear evaluation protocol on the LSFB (3a, 3d) and GSL (3b,3c) dataset based on the number of $k$ shuffled frames at the beginning (from the left to the right (3a ,3b) and at the end (from the right to the left (3c, 3d).

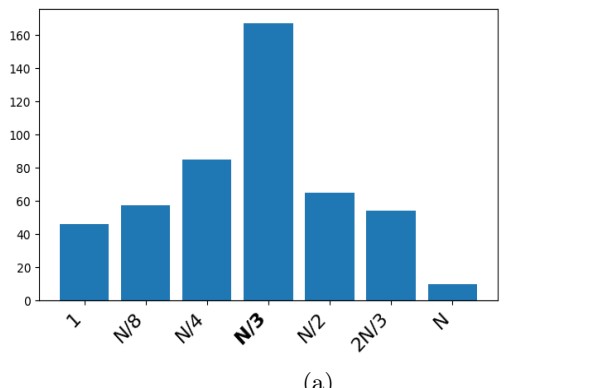
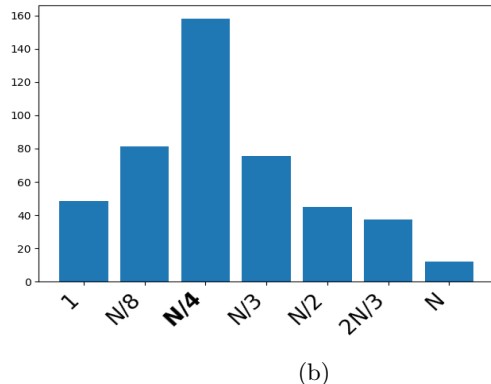

(a)                                               (b)

Figure 4: Histogram of class proportions as a function of permutations starting from the first frames (a) and from the last frames (b).

To give an overview on how Algorithm 1 performs and determine the default $(k_s^*, k_e^*)$ parameters, this part of the paper provides an empirical study of the application of this algorithm on the GSL (medium) and the LSFB (large) datasets. The model parameters are described in section 6.2, and the contrastive method used was SL-FPN trained five consecutive times for each permutation. For computational cost reasons, permutations have been performed on each p frames, with p in $\{1, \frac{N}{8}, \frac{N}{4}, \frac{N}{3}, \frac{N}{2}, \frac{2N}{3}, N\}$.

Figure 3 shows the accuracy (linear evaluation) variations when applying permutations to the first $k_s$ and last $k_e$ frames of a $N$-frame sequence. We observe that when incrementally permuting $k_s$ initial frames, the accuracy evolves progressively until $k_s$ reaches approximately $N/3$. Beyond this threshold (about the first third of the frames), the accuracy begins to decrease continuously, collapsing completely when $k_s = N$ (all

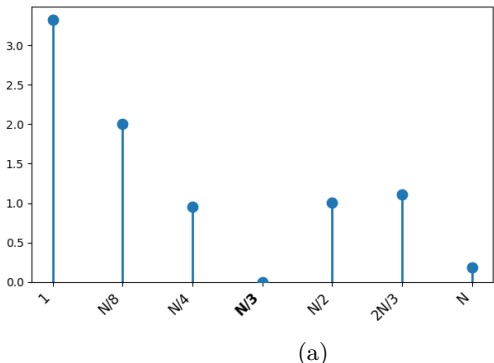 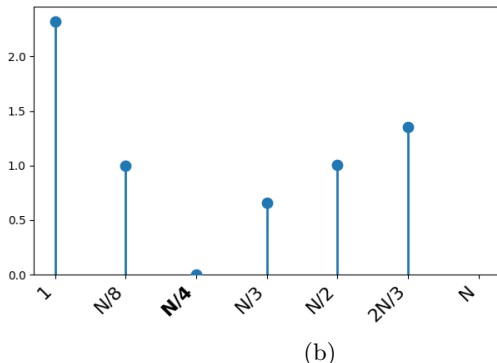

(a)                      (b)

Figure 5: Impact of the global choice ($k_s^*$ and $k_e^*$) on the failure cases.

frames permuted), which seems natural, as all temporal information is then lost. Similarly, when randomly permuting the $k_e$ last frames, the accuracy remains stable until $k_e \approx N/4$ (the last quarter of frames). Beyond this point, accuracy decreases and eventually collapses when $k_e = N$. These results suggest that, the model is robust to temporal order loss in both the first third and last quarter of the sequence. Consequently, frame order appears most critical in the central portion of the sequence (approximately between positions $N/3$ and $N - N/4$). These frames are thus identified as the most useful in the signs identification according to the dataset. After identifying the key frames, the generation of positive pairs is based on applying permutations at different stages. Specifically, permutations are performed on the frames located between the first and $N/3$, as well as on those between $N - N/4$ and the last one.

Choosing parameters that maximize overall accuracy seems like a reasonable approach, but it is important to note that in a sign language dataset, not all signs usually follow the same temporal distribution (i.e., some may become relevant at frame $t$, while others at $t+1$). There exists a global temporal trend in the dataset, as well as trends that may be specific to certain classes or signs in the dataset. Therefore, it is important to measure the impact of choosing $k_s^*$ and $k_e^*$ for the entire dataset. Hence, two further experiments were carried out. The first experiment consists of visualizing, the proportion of observations that achieve their best performance using the parameters $k_s^*$ and $k_e^*$, and those for which the best performances are not achieved with these parameters. The second consists of analyzing the effect of the parameters $k_s^*$ and $k_e^*$ on the signs that do not follow the global temporal distribution of the dataset. The reason behind this experiment is that this study considers the parameters $k_s$ and $k_e$ as those that maximize the overall accuracy. Hence, it is essential to assess the impact of this choice on instances whose optimal parameters do not correspond to $k_s^*$ and $k_e^*$.

For the first experiment, Figure 4 shows two histograms, each representing the proportion of observations as a function of the values of their optimal $k_s$ (first histogram) and $k_e$ (second histogram) on the LSFB dataset. From this figure, we note that generally most observations reach their maximum at $k_s = k_s^*$ and $k_e = k_e^*$. However, we also observe that a subset of instances reaches its maximum at lower values, notably with $k_s = k_e = 1$ or with $k_s \neq k_s^*$ and $k_e \neq k_e^*$. This indicates that a small but non-negligible proportion of signs relies primarily on the initial or the last frames for correct identification. Since the augmentation approach uses the optimal parameters for the global trend in the dataset, this proportion of signs represents the *failure cases* for the proposed augmentation. For these *failure cases*, it is essential to assess the *performance loss* due to the use of these global values instead of the sign-specific optimal ones. This means that, for any signs $s_i$ whose optimal parameters are $(k_s^{s_i} \neq k_s^*, k_e^{s_i} \neq k_e^*)$, the impact of using $(k_s^*, k_e^*)$ in place of $(k_s^{s_i}, k_e^{s_i})$ should be measured. Hence, for each observation $s_i$, we measured the difference between the accuracy obtained with the signs-specific optimal parameters $(k_s^{s_i}, k_e^{s_i})$ and that obtained with the global parameters $(k_s^*, k_e^*)$.

Figure 5 reports the obtained results on the LSFB dataset. From this figure, we note that when $k_s^*$ and $k_e^*$ are not optimal, it incurs a modest loss (1 to 4%) depending on the observations. This loss reaches its maximum value for $k_s = 1$ and $k_e = 1$, which corresponds to signs starting from the first frame and extending to the last frame. These cases represent the main failure cases for the proposed augmentation. We also observe

that as the values of $k_s$ and $k_e$ approach $k_s^*$ and $k_e^*$, the loss becomes less significant. This indicates that the approach is not very sensitive to small variations around the optimal parameters. The fact that most observations reach their maximum at $k_s = k_s^*$ and $k_e = k_e^*$ shows that even though some failure cases may occur, choosing $k_s^*$ and $k_e^*$ as optimal global parameters is justified. To account for these failure cases, new analytical augmentation strategies based on the specific behavior of the observations need to be developed.

Beyond isolated SLR, this augmentation can be apply in other area like continuous SLR. Indeed, in continuous SLR, non-relevant movements (such as coarticulation and repositioning) are also present. In this field, the goal is to minimize the Word Error Rate (WER) (Chen et al., 2022) which is the proportion of signs that are inserted, substituted, or deleted relative to the reference sequence. These coarticulation movements can sometimes be mistakenly interpreted as signs, leading to an increase in WER. By properly calibrating this augmentation method, it can therefore prove to be particularly useful in this domain, as it helps to better handle the natural variations occurring in continuous signing.

Note that, in the proposed augmentation, the parameters $k_s^*$ and $k_e^*$ can be modified according to each specific case and can be done as a preprocessing task. For datasets containing long sign sequences, this procedure can lead to a considerable computational cost. This cost can be estimated as $2 \times k \times T$, where $k$ is the number of tests, $T$ is the execution time per test, and the factor 2 accounts for the test being performed in both directions. To alleviate this issue, one solution is to apply the procedure to subsets of the sequence (subsets consisting of a certain number of frames, as we did in this study), which significantly reduces the number of tests. Another approach is to generalize based on prior information about the data. In our case, since repositioning, coarticulations and non-relevant movements are present across different datasets; the fact that the signers are natives, the parameters $k_s = N/3$ and $k_e^* = N - N/4$ will be used throughout all our experiments. The next section evaluates the SL-FPN, the proposed data augmentation and the SSL-SLR framework (SL-FPN architecture coupled with the proposed augmentation).

---

**Algorithm 1** Search for boundary importance $k_s^*, k_e^*$

---

**Require:** $\mathcal{D}$ (dataset), $\mathcal{C}$ (algorithm), $N$ (sequence length), position $\in \{\text{"first"}, \text{"last"}\}$
**Ensure:** $k_s^*, k_e^*$: the boundary importance
1: **function** FINDOPTIMALK$(\mathcal{D}, \mathcal{C}, N, \text{position})$
2:     $k \leftarrow 1$
3:     $a_{\text{prev}} \leftarrow \text{SegmentEval}(k, \mathcal{D}, \mathcal{C}, \text{position})$
4:     $k \leftarrow 2$
5:     $a_{\text{curr}} \leftarrow \text{SegmentEval}(k, \mathcal{D}, \mathcal{C}, \text{position})$
6:     **while** $k < N$ and $a_{\text{curr}} > a_{\text{prev}}$ **do**
7:         $a_{\text{prev}} \leftarrow a_{\text{curr}}$
8:         $k \leftarrow k + 1$
9:         $a_{\text{curr}} \leftarrow \text{SegmentEval}(k, \mathcal{D}, \mathcal{C}, \text{position})$
10:     **end while**
11:     **return** $k$
12: **end function**
13: $k_s^* \leftarrow$ FINDOPTIMALK$(\mathcal{D}, \mathcal{C}, N, \text{"first"})$
14: $k_e^* \leftarrow$ FINDOPTIMALK$(\mathcal{D}, \mathcal{C}, N, \text{"last"})$
15: **return** $k_s^*, k_e^*$

---

**function** SEGMENTEVAL($k, \mathcal{D}, \mathcal{C}, position$)
    **for** $S = (f_1, \ldots, f_N) \in \mathcal{D}$ **do**
        **if** $position = $ "first" **then**
            $seg_1 \leftarrow \pi_1(f_1, \ldots, f_k)$                                    $\triangleright$ $(f_1, \ldots, f_k)$ are the frames.
            $seg_2 \leftarrow \pi_2(f_1, \ldots, f_k)$
            $S^{(1)} \leftarrow (seg_1, f_{k+1}, \ldots, f_N)$
            $S^{(2)} \leftarrow (seg_2, f_{k+1}, \ldots, f_N)$
        **else**
            $seg_1 \leftarrow \pi_1(f_{N-k+1}, \ldots, f_N)$
            $seg_2 \leftarrow \pi_2(f_{N-k+1}, \ldots, f_N)$                $\triangleright$ $\pi_1, \pi_2$ are two independent random permutations.
            $S^{(1)} \leftarrow (f_1, \ldots, f_{N-k}, seg_1)$
            $S^{(2)} \leftarrow (f_1, \ldots, f_{N-k}, seg_2)$
        **end if**
        Apply $\mathcal{C}$ using $(S^{(1)}, S^{(2)})$ as the positive pair
    **end for**
    **return** accuracy in linear evaluation
**end function**

## 6 Experiments

This section first introduces the datasets used, then provides a quantitative and qualitative evaluation of different methods. Finally, it benchmarks the proposed approach against state-of-the-art methods on different datasets. The quantitative analysis assesses representation quality through performance on a linear evaluation protocol. This evaluation consists of pretraining a contrastive approach, freezing the backbone and train a simple multilayer perceptron on top of it. The qualitative analysis offers a visualization of the latent space and a measure of intra-class inertia for each method. For a fair comparison against the state-of-the-art methods, for datasets where existing methods did not benefit from pretraining (LSFB, LSA, GSL), the SSL-SLR approach was trained from scratch and then fine-tuned. For datasets (WLASL, ASL-Citizen) where state-of-the-art methods were first pretrained, SSL-SLR was pretrained and fine-tuned under the same conditions as the other methods (Wong et al., 2025) and its performances were then evaluated. The obtained results were subsequently compared with those reported in the literature for each dataset. Throughout this entire section, SL-FPN refers to the architecture described in Figure 2 without the proposed augmentation, while SSL-SLR corresponds to the SL-FPN combined with the proposed augmentation (described in Section 5.2).

### 6.1 Datasets and splits

This study uses five datasets with different sizes. These datasets are either made of video clips of signers executing signs independently or sign language sentences with word annotations enabling the extraction of the isolated words. First, we use the French Belgian sign language (LSFB) (Fink et al., 2021). It is one of the largest sign language datasets in the world. It contains 4,567 different classes (signs). Second, the Argentinian sign language (LSA) (Ronchetti et al., 2023) that is a partially recorded dataset in a studio with a uniform background, it contains 64 different classes. Third, the Greek sign language (GSL) (Adaloglou et al., 2021) that is captured in a studio with a consistent background and regulated lighting, it contains 310 classes. Fourth, the American sign language Citizen (ASL Citizen) dataset (Desai et al., 2023) that is a crowdsourced dataset comprising videos of isolated signs performed at home by both native and non-native signers, it contains 2731 different classes. Fifth, the word level American sign language (WLASL) (Li et al., 2020a), a variant of ASL containing 2,000 classes.

For this study, the different datasets were split according to their original papers to ensure fair evaluation and comparison. Some were split 70% for training and 30% for testing (e.g., LSFB), others 80% for training and 20% for testing (e.g., LSA), and 80/10/10 for the GSL. For the state-of-the-art comparison, the 700 most represented classes of the LSFB were used (Fink et al., 2023); for the GSL, ASL Citizen, and LSA, all their classes were used. For the WLASL dataset, we used the WLASL-100 and WLASL-300 subsets, as it is

commonly considered a benchmark for evaluating downstream tasks. For the linear evaluation protocol, the 500 most represented signs of the LSFB and ASL were used.

## 6.2 Training parameters

The experiments use the Python language, typically the PyTorch 2.4 + cu118. For the implementation of the different contrastive learning approaches, we used the PyTorch Lightly library[1], which proposes an implementation for the different methods. Given the sequential nature of data, a transformer encoder, primarily the Vision Transformer (VIT) (Dosovitskiy et al., 2020) provided by Fink et al. (2023) was used as the backbone for the different approaches. As a pre-processing task, the videos were first transformed into skeleton sequences using MediaPipe (Lugaresi et al., 2019). This choice is based on the lower computational cost and insensitivity to visual factors of skeletons compared to videos (Jiang et al., 2021). The batch size was set to 512 signs, each with a maximum sequence length of 64 frames as usual (Desai et al., 2023). During the unsupervised training, the models were trained on 200 epochs, each with parameters specified in their original papers (Chen et al., 2020a; Chen & He, 2021; Grill et al., 2020; He et al., 2020). For the SSL-SLR pretraining, the optimizer was SGD and the learning rate was fixed to 0.001. During the fine-tuning stage, the models were trained for 1000 epochs using SGD optimizer, along with a linear warmup scheduler. The learning rate was progressively increased during the first 600 iterations, then gradually decreased over the remaining 400 (Goyal et al., 2017). The temperature parameter used in the different contrastive learning variants was fixed at 0.5. The transformer-based encoder consisted of 12 blocks, with 8 attention heads and an embedding dimension of 512. We used two layer normalizations at the input embedding stage (Dosovitskiy et al., 2020). The dropout rate was fixed at 0.1. The projection head of all the models was a 2-layer perceptron with a 512-dimensional input and a 128-dimensional output. The predictor $P$ was also a 2-layer perceptron with a ReLU activation function.

## 6.3 Quantitative and time evaluation

For the quantitative evaluation, several types of evaluations were conducted. First, a linear evaluation was conducted that consists of training a classifier on the frozen backbone trained with the unsupervised methods on the training set. For this step, the ASL, LSA, LSFB and GSL were used. To assess the proposed augmentation method, a linear evaluation protocol using the proposed augmentation and the classical augmentations (rotation, Gaussian blur and flip) applied to the whole sequence with all the approaches was conducted. Second, a semi-supervised evaluation with 30% of the annotated data (30% of the training set) was also conducted. For the SLR, where the annotation process is scarce, a semi-supervised evaluation aims to demonstrate the value of learning from unlabeled data to improve model performance when only a small amount of annotated data is available. Third, an evaluation of the possible transfer of the learned representations from a represented sign language to another not represented by the approaches was conducted. For each model, we performed eight consecutive training runs and reported the average accuracy with 95% confidence.

Table 1 presents the results obtained with the proposed SL-FPN compared to other contrastive methods, using first the proposed augmentation method and second standard image augmentation strategies applied to the entire sequence. From this table, we observe that the proposed augmentation helps all the contrastive approaches to better learn meaningful representations. When standard augmentations are applied, performance generally decreases across all methods. This indicates that the proposed data augmentation strategy enables the models to learn more discriminative representations across the different datasets. In some cases, the proposed augmentation yields substantial improvements, with gains of over 6% for the recognition of 500 classes and over 8% for the recognition of 310 classes, highlighting its importance. Therefore, in the following experiments, the proposed augmentation will be consistently applied to compare all the approaches. As the SL-FPN will be coupled with the proposed augmentation, it will be referred to as SSL-SLR.

To evaluate the transferability of learned representations from one sign language to another, we performed unsupervised training on the represented datasets (LSFB and ASL) and conducted linear evaluation on less represented sign language datasets. Table 2 presents the results obtained using different approaches.

---

[1]https://docs.lightly.ai/self-supervised-learning/index.html

Table 1: Linear evaluation protocol with the proposed augmentation method (ours) versus classical image augmentations (rotation, Gaussian blur, flip, translation) applied to all the frames.

| Datasets | SimCLR | MoCo v2 | SimSiam | BYOL | SSL-SLR |
|---|---|---|---|---|---|
| LSFB (our) | $14.16\% \pm 0.24$ | $13.68\% \pm 0.48$ | $15.26\% \pm 0.67$ | $14.72\% \pm 0.65$ | $\mathbf{23.73}\% \pm 0.53$ |
| LSFB | $11.22\% \pm 1.04$ | $10.91\% \pm 1.86$ | $11.84\% \pm 1.71$ | $11.15\% \pm 1.12$ | $\mathbf{16.07}\% \pm 1.41$ |
| ASL (our) | $14.13\% \pm 0.42$ | $14.69\% \pm 0.39$ | $15.91\% \pm 0.56$ | $16.43\% \pm 0.96$ | $\mathbf{20.46}\% \pm 1.21$ |
| ASL | $11.04\% \pm 0.69$ | $11.09\% \pm 1.12$ | $12.17\% \pm 1.36$ | $12.78\% \pm 0.78$ | $\mathbf{15.43}\% \pm 1.20$ |
| GSL (our) | $34.19\% \pm 0.85$ | $36.15\% \pm 0.69$ | $32.01\% \pm 0.54$ | $34.09\% \pm 0.93$ | $\mathbf{47.76}\% \pm 0.79$ |
| GSL | $31.09\% \pm 1.75$ | $30.15\% \pm 1.56$ | $30.11\% \pm 1.47$ | $30.89\% \pm 1.11$ | $\mathbf{35.13}\% \pm 1.34$ |
| LSA (our) | $34.02\% \pm 1.24$ | $35.69\% \pm 1.06$ | $30.06\% \pm 2.14$ | $37.47\% \pm 1.51$ | $\mathbf{41.74}\% \pm 1.08$ |
| LSA | $29.10\% \pm 2.89$ | $27.17\% \pm 2.76$ | $26.16\% \pm 2.46$ | $32.37\% \pm 2.61$ | $\mathbf{36.71}\% \pm 1.97$ |

Table 2: Transferring from one sign language to another with the proposed augmentation method.

| Datasets | SimCLR | MoCo v2 | SimSiam | BYOL | SSL-SLR |
|---|---|---|---|---|---|
| LSFB to LSA | $33.40\% \pm 1.16$ | $32.47\% \pm 1.24$ | $31.15\% \pm 0.67$ | $35.67\% \pm 0.47$ | $\mathbf{46.41}\% \pm 0.89$ |
| LSFB to GSL | $33.24\% \pm 0.68$ | $35.24\% \pm 0.70$ | $40.24\% \pm 0.44$ | $34.22\% \pm 0.51$ | $\mathbf{54.78}\% \pm 0.56$ |
| ASL to LSA | $34.74\% \pm 0.47$ | $31.53\% \pm 0.36$ | $39.16\% \pm 0.39$ | $38.96\% \pm 0.63$ | $\mathbf{43.84}\% \pm 0.37$ |
| ASL to GSL | $32.46\% \pm 0.43$ | $33.58\% \pm 0.43$ | $35.25\% \pm 0.65$ | $34.89\% \pm 1.45$ | $\mathbf{39.73}\% \pm 1.21$ |

The SSL-SLR method clearly outperforms the others. This result suggests that the representations learned by SSL-SLR can also be more effectively transferable from one sign language to another compared to the representations learned by approaches like SimCLR, MoCo, BYOL and SimSiam.

The importance of representations learned from unannotated sign language was also evaluated in low-resource scenarios. To simulate such settings, a fine-tuning using only 30% of the training set is conducted. The corresponding results are presented in Table 3. The proposed SSL-SLR consistently outperforms standard contrastive methods. This highlights the robustness and transferability of the representations learned through our method, even when limited labeled data is available.

The above results demonstrate the effectiveness of the proposed approach for SLR tasks compared to existing contrastive and self-supervised methods. They also highlight the impact that contrastive learning can have on reducing reliance on annotations in SLR. This advancement represents a significant contribution to the development of models that offer acceptable performance while alleviating the tedious and costly process of manual annotation.

To better position the SL-FPN approach within the self-supervised literature, it is essential to have an idea of the execution time during contrastive learning compared to other approaches such as BYOL and SimSiam, which also do not use negative pairs. For this purpose, the execution time of the self-supervised training for the SL-FPN, SimSiam, and BYOL approaches has been reported in Table 4. On the different datasets, we observe that SimSiam achieves the best execution time. This seems natural as it employs a single encoder without requiring negative pairs. In contrast, BYOL demonstrates longer execution times compared to both SL-FPN and SimSiam. This can be explained by the fact that it employs two distinct encoders, and one is updated using an exponential moving average. SL-FPN approach shows a slightly higher execution time than SimSiam, but lower than BYOL. This increase is due to the use of the original instance. The fact that it does not require performing augmentation on the original instance reduces moderate considerably its execution time. Hence, despite its effectiveness, SL-FPN remains reasonable in terms of execution time.

### 6.3.1 Qualitative Evaluation

The qualitative evaluation consists in general of visualizing the embedding space to evaluate at which points the instances of the same class are close in the space. So, 2D visualization (McInnes et al., 2018) was used to show the embedding space. Ideally, the signs with the same class should be closer in the embedding space.

Table 3: Fine-tuning with 30% of annotations with our augmentation.

| Datasets | SimCLR | MoCo v2 | SimSiam | BYOL | SSL-SLR |
|---|---|---|---|---|---|
| LSFB | $42.69\% \pm 3.36$ | $42.23\% \pm 2.14$ | $43.69\% \pm 2.69$ | $41.40\% \pm 2.04$ | $\mathbf{49.93}\% \pm 2.98$ |
| ASL | $47.43\% \pm 0.77$ | $47.49\% \pm 0.54$ | $47.23\% \pm 0.63$ | $47.02\% \pm 0.51$ | $\mathbf{49.28}\% \pm 0.79$ |
| GSL | $78.82\% \pm 2.96$ | $77.42\% \pm 2.87$ | $77.02\% \pm 2.95$ | $78.04\% \pm 2.65$ | $\mathbf{83.86}\% \pm 2.01$ |
| LSA | $87.69\% \pm 1.48$ | $88.04\% \pm 1.69$ | $87.96\% \pm 1.36$ | $88.64\% \pm 1.36$ | $\mathbf{92.76}\% \pm 1.63$ |

Table 4: Linear evaluation time in second by the different approaches.

| Dataset | SimSiam | BYOL | SL-FPN |
|---|---|---|---|
| LSFB | $\mathbf{2.49} \times 10^4$ | $2.67 \times 10^4$ | $2.56 \times 10^4$ |
| GSL | $\mathbf{9.77} \times 10^3$ | $1.04 \times 10^4$ | $1.02 \times 10^4$ |
| LSA | $\mathbf{2.67} \times 10^3$ | $2.86 \times 10^3$ | $2.77 \times 10^3$ |
| ASL | $\mathbf{2.22} \times 10^4$ | $2.46 \times 10^4$ | $2.43 \times 10^4$ |

Figure 6 illustrates the 2D visualization of embedding spaces. Given the high number of classes, we randomly select and show the different instances of a class across the datasets on the embedding space. On the ASL, GSL, and LSA datasets, for the randomly chosen class (sign), the embeddings produced by SSL-SLR are closer than those from the SimCLR, MoCo, BYOL, and SimSiam approaches. This demonstrates the effectiveness of the proposed method in generating higher-quality representations compared to the other approaches. However, on the LSFB dataset, the instances are scattered throughout the embedding space across the different approaches. This can be attributed to the complexity of this dataset (many signs are visually similar, several variants of the same signs, a large number of classes). Additionally to this visualization, an intra-class inertia was computed for the different dataset. Table 5 presents the intra-class inertia obtained by the different approaches on several datasets. It is observable from this table that the proposed approach performs better than the others.

## 6.4 Comparison Against State-of-the-art Methods

To demonstrate the effectiveness of the proposed SSL-SLR in the SLR literature, a benchmark and a comparison against others state-of-the-art methods on different datasets are conducted. As in the literature (Hu et al., 2021; 2023; Jiang et al., 2024; Zhao et al., 2023), SSL-SLR was first pre-trained and fine-tuned for the comparison. The goal is to compare the proposed approach with previous works done in terms of accuracy during the recognition task. Table 6 presents the obtained results, the best results are in bold. From this table, it is observable that, on the LSFB, LSA, ASL Citizen and the GSL dataset, the proposed approach achieved several well-known state-of-the-art methods. On the WLASL, the proposed SSL-SLR surpassed several of them in top-5 accuracy.

## 7 Ablation Study

We examined the impact of the predictor equipped with the stop-gradient operator, the use of the original sample, layer normalization, and the permutation of the components of the proposed SL-FPN during training. We used the LSFB dataset due to its size and diversity. Feature collapse occurs when the model produces the same representations for different signs. To determine when a model collapses, (Chen & He, 2021) showed that, given $d$ the embedding size within the interval $[0, \frac{1}{\sqrt{d}}]$, the standard deviation of the model drops to zero. Following their protocol, we examine the model output in the same way. To investigate this, we trained the model in four different settings: (i) without the predictor, the stop-gradient and layer normalization (without (p, $s_t$ & LN)); (ii) without the predictor, the stop-gradient and with layer normalization (without p, $s_t$ & with LN); (iii) without the original sample (without o); and (iv) by permuting the original sample and one of the positive pairs (perm) in the proposed SL-FPN. Figure 7 shows the results obtained. When both the predictor and the layer normalization are used, there is no sign of feature collapse. However, when both are removed, the standard deviation drops to zero after a few training steps, clearly indicating collapse.

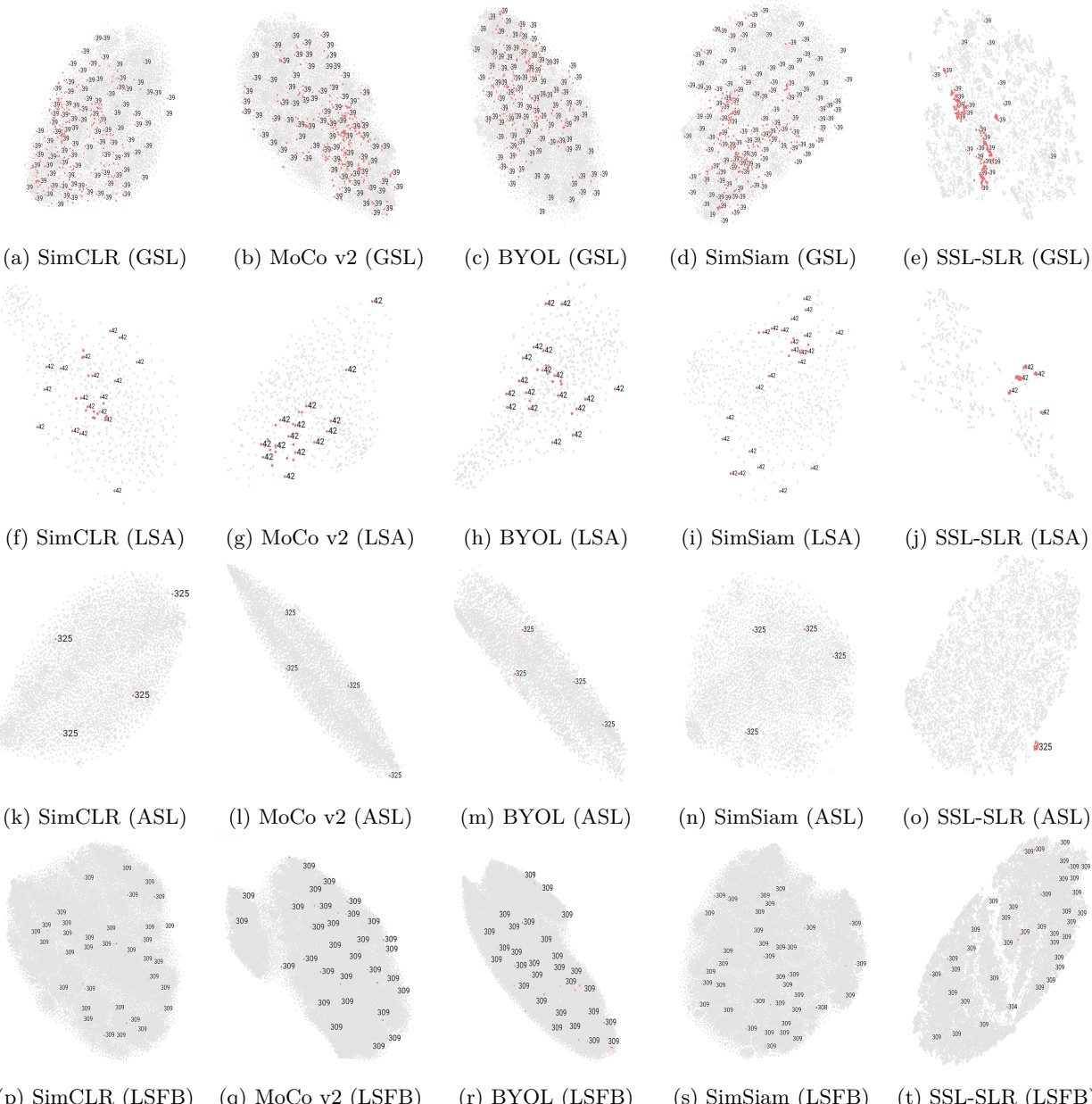

Figure 6: Visualization of the embeddings using SimCLR, MoCo, BYOL, SimSiam, and the proposed method on four datasets: GSL, LSA , ASL and LSFB.

When only layer normalization is used, the standard deviation gradually decreases, which suggests that it slows down the collapse but does not fully prevent. We also note that during the training, when the original sample is not used and by permuting the order of the inputs in the proposed SL-FPN, the accuracy decreases significantly. This suggests that using the original input, when it is well-calibrated, can significantly improve the quality of the representations.

Table 5: Intra-class inertia of different approaches

| Datasets | SimCLR | MoCo v2 | SimSiam | BYOL | SSL-SLR (Ours) |
|---|---|---|---|---|---|
| LSFB | $1.86 \times 10^4$ | $1.84 \times 10^4$ | $1.36 \times 10^4$ | $1.84 \times 10^4$ | $\mathbf{0.57} \times 10^4$ |
| ASL | $3.11 \times 10^3$ | $1.37 \times 10^3$ | $2.81 \times 10^3$ | $1.54 \times 10^3$ | $\mathbf{0.41} \times 10^3$ |
| GSL | $1.25 \times 10^4$ | $0.91 \times 10^4$ | $0.69 \times 10^4$ | $0.67 \times 10^4$ | $\mathbf{0.17} \times 10^4$ |
| LSA | $3.45 \times 10^2$ | $2.72 \times 10^2$ | $2.27 \times 10^2$ | $1.22 \times 10^2$ | $\mathbf{0.25} \times 10^2$ |

Table 6: Top-1 and Top-5 accuracy of state-of-the-art methods and the proposed SSL-SLR on various sign language datasets.

| Dataset | Method | Top-1 (%) | Top-5 (%) |
|---|---|---|---|
| ASL Citizen | SignCLIP (Jiang et al., 2024) | 46.00 | 77.00 |
|  | SignRep (avg) (Wong et al., 2025) | 37.47 | 68.77 |
|  | SignRep (weighted) (Wong et al., 2025) | **49.95** | **80.09** |
|  | SSL-SLR (Ours) | 47.06 | 78.96 |
| LSFB | Fink et al. (2023) | 54.40 | - |
|  | SSL-SLR (Ours) | **56.81** | - |
| LSA | Masood et al. (2018) | 95.21 | - |
|  | Alyami et al. (2024) | 98.25 | - |
|  | SSL-SLR (Ours) | **99.07** | - |
| GSL | Adaloglou et al. (2021) | 89.74 | - |
|  | Papadimitriou et al. (2024) | 96.25 | - |
|  | SSL-SLR (Ours) | **96.73** | - |
| WLASL-100 | PSLR (Tunga et al., 2021) | 60.15 | 83.98 |
|  | SignBERT (Hu et al., 2021) | 76.36 | 91.09 |
|  | BEST (Zhao et al., 2023) | 77.91 | 91.47 |
|  | SignBERT+ (Hu et al., 2023) | **79.84** | 92.64 |
|  | SSL-SLR (Ours) | 77.95 | **93.02** |
| WLASL-300 | PSLR (Tunga et al., 2021) | 42.18 | 71.71 |
|  | SignBERT (Hu et al., 2021) | 62.72 | 85.18 |
|  | BEST (Zhao et al., 2023) | 67.66 | 89.22 |
|  | SignBERT+ (Hu et al., 2023) | **73.20** | 90.42 |
|  | SSL-SLR (Ours) | 71.21 | **90.74** |

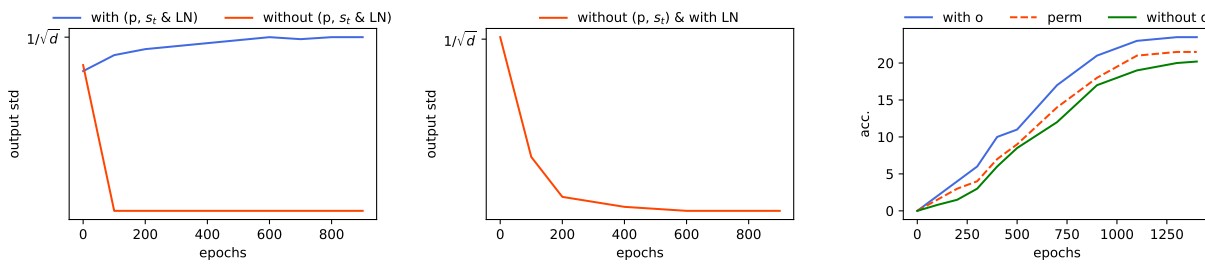

Figure 7: Standard deviation and linear evaluation accuracy of the model with ablation of predictor, stop-gradient, original sample and layer normalization on the LSFB dataset.

## 8    Conclusion

This paper introduces a new self-supervised framework for sign language recognition. The framework is based on a novel self-supervised approach and a new augmentation designed to degrade the non-informative part of a sign. The self-supervised approach leverages both positive pairs and the original instance. It is designed to be invariant to non-informative parts of signs while producing similar representations for a sign and its augmented variants. To validate the effectiveness of the proposed approach, several evaluation protocols, including linear evaluation, cross-lingual representation transfer and semi-supervised learning have been conducted. The results demonstrate the effectiveness of the proposed method compared to existing contrastive and state-of-the-art approaches across these different settings. This advancement represents a significant contribution to the development of models that offer acceptable performance while alleviating the tedious and costly process of manual annotation. Despite its effectiveness, the performance still has room for improvement on large-scale datasets. Furthermore, the proposed method to determine the boundary importance is currently determined empirically. In future work, we plan to develop a non-empirical method to determine boundary importance. This will enable to mitigate the failure cases that face the current version. We also plan to extend this work for the cases like continuous sign language.

## 9    Acknowledgments

The present research benefited from computational resources made available on Lucia, the Tier-1 super-computer of the Walloon Region, infrastructure funded by the Walloon Region under the grant agreement n°1910247. The authors thank Valentin Delchevalerie and Simon Lejoly for their insightful comments and feedback. They also thank the TMLR reviewers and the action editor for their constructive remarks on the manuscript.

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
