# OpenReview forum: "SSL-SLR: Self-Supervised Representation Learning for Sign Language Recognition"
_TMLR — Accepted by TMLR_

### Review · Reviewer_uxhV · 2025-11-18

**Summary Of Contributions:**

The paper introduces a self-supervised method for sign recognition that is composed of two parts:
- a three-loss architecture that tries to map positive augmented samples close to the original sample in the latent space, followed by a predictor for classification
- an augmentation scheme that relies on a contrastive method to determine the most important frames of the sample that should be augmented

I am not an expert in the field, so I cannot comment properly on the body of literature cited; however citations seem abundant especially as regards baselines the method is compared to.

**Additional Comments:**

I think this work should be considered for a TMLR certification.

**Audience:**

Yes

**Audience Explanation:**

Avoiding the need to craft negative samples takes a huge burden off practitioners, which makes the method more appealing to a broader audience. The topic is in line with TMLR's scope, and it is well executed.

I think this paper should be accepted following minor adjustments mentioned below.

**Broader Impact Concerns:**

I have no specific concerns.

**Claims And Evidence:**

Yes

**Claims Explanation:**

This paper is well-written, organized, and with a reader-oriented style that makes it easy to read and understand the new contributions. The authors have justified every claim made in the paper with either empirical evidence or by supporting citations. It was a pleasure to read it.

I believe the claims of better performance are properly supported. I specifically appreciate the intuition behind the use of a contrastive method to detect the best parts to augment, which exploits the inductive bias of certain architectures (Transformer) to achieve the intended goal.

**Requested Changes:**

I would like the authors to further clarify certain aspects:
- Sec 5.1 (4th line): contrastive methods do not only use positive pairs. I think the sentence might be ambiguously written
- Sec 5.1: "randomly applying" -> please specify in which sense: do you pick at random between T and T' ?
- Please discuss why you do not utilize a specular loss as L2 using the other augmentation
- End of page 4: n should be $n$
- Equation 1: please provide an intuitive understanding for L3
- Sec 5.2: "it is well-established that..." please provide a reference supporting your statement
- I believe the authors should discuss the limitations of using a pre-trained contrastive method that is not necessarily readily available. Actually it was not clear if the contrastive method was trained or not. What is the trade-off here?
- Please improve figures by adding labels (e.g. Fig 3) as well as not using screenshots (Fig 6)
- Sec 5.2 "the accuracy evolves": does it increase/decrease? why do you think it should increase as in Figure 3 when you apply permutations of the first/last elements? This is not intuitive to me
- Please clarify what "accuracy in linear evaluation" means. I could not find a proper explanation in the paper.
- Page 9: there is a 80/10/10 split reported: how did you use the validation set?
- I think the authors should discuss why they did not perform hyper-parameter tuning, including whether hyper-parameter tuning was performed for other methods. If no method performed a proper tuning of the hyper-parameters, it might happen that the ranking of the various models is biased by a certain architectural choice.

---

> ### Author Response · Authors · 2025-12-28
>
> Thank you for taking the time to review our manuscript. We are very pleased that you appreciated our work and even suggested accepting it with minor revisions. Your comments have been extremely helpful in improving both the quality and the clarity of the manuscript. In the current version, we have carefully addressed them and marked them in blue. Below is a non-exhaustive list of our responses to your concerns
>
> The sentence related to *contrastive learning*, which was potentially confusing, has been rewritten.
> We have clarified this point and explained more precisely that `T` and `T'` correspond to two distinct augmentations, selected from the set of transformations used during training.
>
> Typographical errors, such as the use of `n` instead of `𝑛` where appropriate, have been corrected. The intuition behind the use of the different *losses* ( particularly L3) has also been better articulated.
>
> Necessary references to support certain points (particularly the link between the quality of the learned representations and the performance of contrastive models) have been added.
>
> The quality of Figure 6 (7 in the current version) has been improved: we no longer use screenshots in this version.
> .
> Regarding the labels in Figure 3, since the content of the figure is already described in the caption, we opted not to repeat this information to avoid redundancy and to prevent visually overloading the figure. All essential information is therefore provided directly in the caption. Additionally, we have included a brief explanation of what *accuracy* entails in linear evaluation.
>
> Here are some answers to other concerns:
>
> **Sec 5.2 "the accuracy evolves: does it increase/decrease?, [...]"**
>
> We would like to thank the reviewer for raising this concern. Regarding the evolution of accuracy, we observe an initial increase followed by a decrease. This behavior can be explained by the fact that not all parts of a sign video are equally relevant for classification. As the model learns to become invariant to irrelevant parts, it progressively focuses its attention on the truly informative parts, which leads to improved performance. However, once the perturbations start affecting the relevant parts, the performance naturally begins to drop. To provide intuition, we can draw an analogy with image classification, where the background is often irrelevant for identifying the Region of Interest. The more the model becomes invariant to the background, the more accurate and efficient it becomes at identifying the RoI. In the current version of the manuscript, we have provided the intuition behind this behavior.
>
> **I think the authors should discuss why they did not perform hyper-parameter, [...]**
>
> We would like to thank the reviewers for this remark. For the comparison with methods from the literature, we used the results reported in the original papers. For the re-implemented methods, certain hyperparameters were fixed identically across all approaches, in particular the batch size. This was chosen to be sufficiently large given the available computational resources (e.g., a batch size of 512) in order to meet the requirements of contrastive methods such as SimCLR and MoCo. Regarding method-specific hyperparameters (e.g., the memory size in MoCo), we used the values recommended in the original works, with the aim of remaining as faithful as possible to the experimental configurations proposed by the authors.

---

### Review · Reviewer_b4zV · 2025-12-13

**Summary Of Contributions:**

Summary:

This paper introduces SSL-SLR, a self-supervised learning framework for sign language recognition that addresses two key limitations of existing contrastive methods: the high semantic similarity of negative pairs in SLR and the presence of non-discriminative temporal segments within sign videos. The framework combines a negative-free three-branch architecture (SL-FPN) that jointly leverages augmented views and the original sample, with a boundary-aware temporal augmentation strategy that degrades non-informative frames. Experiments on multiple sign language datasets show consistent improvements over standard contrastive and self-supervised baselines under linear probing, semi-supervised learning, and cross-lingual transfer.

Strengths:

1. The paper provides a clear and accurate diagnosis of why standard instance-discrimination methods (e.g., SimCLR, MoCo v2) underperform in sign language recognition, where different signs often share similar hand shapes and motion patterns, motivating the move away from explicit negative pairs.

2. The proposed SL-FPN architecture is simple yet well motivated, distinguishing itself from BYOL and SimSiam by explicitly incorporating the original sample alongside augmented views, thereby improving the utilization of training signals within a single-encoder, negative-free framework.

3. The boundary-aware temporal augmentation strategy is a strong domain-specific contribution, effectively addressing coarticulation and repositioning effects by identifying and degrading non-discriminative temporal segments using transformer sensitivity to temporal order.

4. The experimental evaluation is thorough and well designed, covering multiple datasets (LSFB, ASL, GSL, and etc.) and learning settings, including linear probing, semi-supervised learning, and cross-lingual transfer, with consistent performance gains.


Weaknesses:

1. The positioning relative to recent self-supervised SLR-specific methods (e.g., SignRep) is limited, and a clearer comparison would help disentangle the benefits of the proposed augmentation strategy from architectural choices.

2. The boundary importance selection relies on empirical ratios (N/3, N/4) derived from observed trends, and its robustness across datasets with different signing styles or continuous signing settings remains unclear.

3. The computational cost of the boundary estimation procedure is not discussed in detail, and its scalability to longer sequences or larger datasets would benefit from further clarification.

**Audience:**

Yes

**Audience Explanation:**

The paper directly addresses representation learning under limited supervision, a topic of broad interest to the TMLR community. The paper will be of interest to researchers working on self-supervised learning, video representation learning, and low-resource sequence modeling. In particular, the domain-aware augmentation strategy and the analysis of why standard contrastive assumptions fail in SLR provide insights that extend beyond sign language recognition.

**Claims And Evidence:**

Yes

**Claims Explanation:**

The claims are well supported by comprehensive empirical evidence across multiple datasets and evaluation settings.

1. Comparisons with standard contrastive and negative-free baselines, together with ablation studies, demonstrate the effectiveness of the proposed SL-FPN architecture and boundary-aware temporal augmentation.
2. Both quantitative results (linear probing, semi-supervised learning, cross-lingual transfer) and qualitative analyses (embedding visualization, intra-class inertia) support the claimed improvements in representation quality.

**Requested Changes:**

1. Add a related work section would benefit from a more explicit discussion of recent self-supervised methods tailored for sign language representation learning, such as SignRep, and other emerging approaches that focus on sign-specific inductive biases rather than generic visual SSL. Including these works would help clarify how SSL-SLR differs from existing SLR-focused representation learning methods, particularly in terms of boundary-aware augmentation versus architectural design.

2. While the paper positions SL-FPN as a negative-free approach distinct from BYOL and SimSiam, the conceptual differences could be articulated more clearly. Discuss why incorporating the original (non-augmented) sample provides additional benefits beyond existing asymmetric Siamese frameworks, and under what conditions this design choice is expected to be most effective.

3. The boundary importance parameters(N/3, N/4) are empirically derived from selected datasets(GSL, LSFB). The paper would benefit from a more detailed discussion on how sensitive performance is to deviations from these ratios, whether similar trends are observed across various datasets with different frame rates or signing styles, and how this procedure might generalize to continuous or sentence-level sign language recognition.

4. The current experiments focus on isolated sign recognition. A brief discussion on how the proposed SL-FPN architecture and boundary-aware augmentation might extend to continuous signing, sentence-level SLR, or sign language translation pipelines would strengthen the broader relevance of the work.

---

> ### Author Response · Authors · 2025-12-28
>
> Thank you very much for the time you spent reading and evaluating our manuscript. We are pleased to learn that you appreciated the proposed SL-FPN architecture, the data augmentation strategy, and the reported results. Your comments have been extremely helpful in improving both the quality and the clarity of the manuscript. In the current version, the updated points are marked in blue.
>
> **Add a related work section would benefit from a more explicit discussion of recent self-supervised methods, [...]**
>
> We acknowledge that the first version of the manuscript did not reference certain unsupervised approaches, such as SignRep. In the revised version, we have included SignRep in Section 3. Additionally, in the state-of-the-art comparison (Section 6.4), we have added an evaluation of SignRep on common benchmark datasets to ensure a fair and meaningful comparison.
>
> **While the paper positions SL-FPN as a negative-free approach distinct, [...]**
>
> In the current version of the manuscript, the conceptual differences and the benefits of incorporating the original instance are explicitly discussed. We also show that in situations involving semantic inconsistency (a well-known problem in self-supervised learning), the SL-FPN framework with the original instance can be particularly effective.
>
> **The boundary importance parameters (N/3, N/4) are empirically derived from selected datasets (GSL, LSFB). The paper would benefit from a more detailed discussion on how sensitive performance is to deviations from these ratios, [...]**
>
> In the current version of the manuscript (Section 5.2), we analyzed the proportion of best-performing configurations per sign as a function of different values of  `k_s` and `k_e`. Based on this analysis, we provided a detailed discussion of the method’s sensitivity to variations in these parameters and failure cases.
>
> **The current experiments focus on isolated sign recognition. A brief discussion, [...]**
>
> Thank you for your comment regarding continuous recognition. We have added a discussion about this point; particularly, we have shown that the proposed augmentation can be extended to other areas (Section 5.2), such as continuous SLR.
> In continuous SLR, even between signs, coarticulation and non-pertinent movements are present. A model may incorrectly classify these movements as valid signs, which directly leads to an increase in the word error rate (WER). We briefly explain how our approach could be beneficial in such scenarios.

---

### Review · Reviewer_hRW7 · 2025-12-14

**Summary Of Contributions:**

The paper proposes SSL-SLR, a self-supervised learning framework designed specifically to address the scarcity of annotated data in Sign Language Recognition (SLR). The contributions are twofold:

SL-FPN Architecture: A novel "negative-free" self-supervised approach that utilizes three branches: the original sample and two augmented versions. It minimizes the distance between these representations and uses a predictor with a stop-gradient operator to align the original sample with its augmented variants, eliminating the need for negative pairs or clustering mechanisms.

Temporal Augmentation Strategy: A new data augmentation technique that empirically identifies "boundary importance" (frames $k_{s}^{*}$ and $k_{e}^{*}$). The method degrades non-relevant parts of the video (typically the start and end) via permutation, forcing the model to focus on the discriminative "internal" segments of the sign.

**Audience:**

Yes

**Audience Explanation:**

SLR Researchers will be interested in the data efficiency of the method, particularly the ability to achieve competitive results with only 30% of annotated data and the improved transferability between sign languages.

The SL-FPN architecture is interesting as a simplification of existing SSL methods (like BYOL/SimSiam) by integrating the "original" un-augmented sample directly into the loss function to prevent collapse and improve alignment.

**Broader Impact Concerns:**

Bias in Heuristics: There is a minor concern regarding the "boundary importance" heuristic18. If the timing of signs ($k_{s}^{*}$ and $k_{e}^{*}$) varies significantly between native signers and learners (or across different demographics), hard-coding these thresholds could introduce bias, optimizing the model for a specific style of signing. Adding a brief statement on the demographic diversity of the signers used to derive these thresholds would address this.

**Claims And Evidence:**

Yes

**Claims Explanation:**

The claims are supported by experimentation, though with a reliance on heuristic derivation for the augmentation strategy.

Empirical Results: The authors provide comprehensive benchmarks across five datasets (LSFB, LSA, GSL, ASL Citizen, WLASL)3. The results show that SSL-SLR consistently outperforms standard contrastive baselines (SimCLR, MoCo v2, BYOL, SimSiam) in linear evaluation, semi-supervised learning (with only 30% labels), and cross-lingual transfer tasks.

Visual Confirmation: The UMAP visualizations and intra-class inertia metrics convincingly demonstrate that SSL-SLR produces more compact and discriminative embedding spaces than competing methods.

Heuristic Limitation: The determination of "relevant frames" is derived from aggregate statistics (Fig. 3), leading to a global rule of preserving the middle segment (between $N/3$ and $N-N/4$). While shown to be effective on average, the claim that this method accurately identifies relevant parts for all signs is not fully substantiated for outliers that may have different temporal dynamics.

**Requested Changes:**

Robustness to Skeleton Noise: The method relies on MediaPipe for skeleton extraction. It would be beneficial to discuss if the "degradation" via permutation affects the model's robustness to skeleton estimation errors, which are common in sign language videos.

Clarification of Heuristic Generalization: The paper applies a fixed temporal crop ($N/3$ and $N/4$) across all datasets based on the "average" performance shown in Figure 3. The authors must address or analyze the failure cases where specific signs do not follow this temporal distribution (e.g., signs that start immediately without a setup phase). A critical discussion on the variance of $k_{s}^{*}$ and $k_{e}^{*}$ across different sign classes is needed.

Computational Cost Analysis: While the method eliminates negative pairs (reducing batch complexity), it adds a third branch for the original sample. A comparison of training time and memory usage against SimSiam/BYOL would strengthen the efficiency claims.

---

> ### Author Response · Authors · 2025-12-28
>
> Thank you very much for the time you took to read our manuscript. We are pleased to see that you found the proposed SL-FPN architecture interesting and that you consider the work relevant for a community such as TMLR. We greatly appreciate your constructive feedback, and we have carefully taken your concerns into account while revising the manuscript accordingly. In the current version, the updated points are marked in blue. Below are some responses to your concerns and the ways we have addressed them.
>
> **Robustness to Skeleton Noise, […]**
>
> As you mention, our method relies on MediaPipe for skeleton extraction, which may introduce estimation errors, typically localized at the joint level (e.g., due to occlusions or tracking failures). In our setting, skeleton extraction is treated as a preprocessing step, and the resulting sequences are assumed to be fixed during training. Since the proposed augmentation does not alter the local frame-level distribution, it does not affect the model’s robustness to skeleton estimation errors. To improve the clarity, we have added a brief note in Section 5.2.
>
> **Clarification of Heuristic Generalization, [...]**
>
> For this study, the boundary importance parameters `k_s*` and `k_e*` were empirically selected based on the overall performance observed across the evaluated datasets. We thank the reviewer for this insightful remark, which helped improve the quality of the paper.   To address your concerns, we conducted a class-wise analysis of the performance with respect to the parameters `k_s*` and `k_e*`. Specifically, we analyzed the proportion of best-performing configurations per signs as a function of different values of `k_s` and `k_e`. Based on this analysis, we analyze the failure cases and therefore evaluated the impact of the choice of  `k_s*` and `k_e*` on this failure cases. This analysis was carried out in Section 5.2.
>
> **Computational costs, [...]**
>
> The proposed SL-FPN architecture involves three forward passes through a shared encoder per sample. The additional branch corresponds to the original sequence, which is processed by the same encoder without introducing extra augmentations. This results in a moderate increase in computational cost (i.e., 2 minutes on the LSA dataset, 10 minutes on LSFB, and 6 minutes on GSL) compared to SimSiam and less than BYOL that requires two encoders. In the current version of the manuscript (Section 6.3), we have added a computational cost analysis.
>
> **Bias in Heuristics: There is a minor concern regarding, [...]**
>
> As recommended, a brief statement on the demographic diversity of the signers has been introduced in  Section 5.2.

---

### Decision · Action_Editor_zoiN · 2026-02-14

**Recommendation:** Accept as is

**Additional Comments:**

This paper presents a well-executed self-supervised framework for sign language recognition. The contributions are clearly motivated, empirically validated across multiple datasets, and address practical limitations of standard contrastive methods. Overall, the work is well supported by empirical evidence, relevant to TMLR’s audience, and suitable for acceptance.

**Audience:**

Yes

**Audience Explanation:**

The work addresses a practical and widely relevant problem in self-supervised representation learning for video sequences under limited supervision. TMLR readers interested in SSL, video representation learning, low-resource sequence modeling, and domain-specific augmentations will find the findings valuable.

**Claims And Evidence:**

Yes

**Claims Explanation:**

The authors provide extensive empirical evidence across multiple sign language datasets demonstrating that SSL-SLR outperforms standard contrastive and negative-free self-supervised baselines. The visualizations and intra-class inertia analyses support the claim that the method learns more compact and discriminative embeddings. Computational cost and robustness to skeleton extraction noise are also clarified, providing a convincing and transparent picture of the method’s effectiveness.